# Changes in the Oswestry Disability Index after a 3-Week In-Patient Multidisciplinary Body Weight Reduction Program in Adults with Obesity

**DOI:** 10.3390/jcm11113175

**Published:** 2022-06-02

**Authors:** Munkh-Erdene Bayartai, Hannu Luomajoki, Roberta De Micheli, Gabriella Tringali, Nicoletta Marazzi, Alessandro Sartorio

**Affiliations:** 1Institute of Physiotherapy, School of Health Professions, Zurich University of Applied Sciences, 8400 Winterthur, Switzerland; luom@zhaw.ch; 2Department of Physical Therapy, School of Nursing, Mongolian National University of Medical Sciences, Ulaanbaatar 14210, Mongolia; 3Experimental Laboratory for Auxo-Endocrinological Research, Istituto Auxologico Italiano, Istituto di Ricovero e Cura a Carattere Scientifico (IRCCS), 28824 Piancavallo (VB), Italy; r.demicheli@auxologico.it (R.D.M.); g.tringali@auxologico.it (G.T.); n.marazzi@auxologico.it (N.M.); sartorio@auxologico.it (A.S.); 4Division of Auxology & Metabolic Diseases, Istituto Auxologico Italiano, Istituto di Ricovero e Cura a Carattere Scientifico (IRCCS), 28824 Piancavallo (VB), Italy

**Keywords:** obesity, adulthood, disability, rehabilitation

## Abstract

The aim of this study was to examine the short-term changes in disability after an inpatient, multidisciplinary body weight reduction program (BWRP) in adults with obesity. A total of 160 individuals (males: 52, females: 108, BMI > 35 kg/m^2^) hospitalized for a 3-week multidisciplinary BWRP were recruited into the study. Body composition, lower limb muscle power, fatigue severity, and disability were measured at the beginning and end of the intervention by means of bioimpedance analysis, a stair climbing test (SCT), the Fatigue Severity Scale (FSS), and the Oswestry disability index (ODI), respectively. At the end of the 3-week BWRP, an average body weight reduction of 5.0 kg (CI 95% −5.3; −4.6, *p* < 0.001) was determined, as well as an improvement in all parameters measured. Clinically meaningful reductions in disability were observed in the moderate disability (Δ = −11.8% CI 95% −14.3; −9.3, *p* < 0.001) and severe disability (Δ = −15.9% CI 95% −19.6; −12.2, *p* < 0.001) groups. Reductions in disability were explained only by improvements in the SCT (Δ = −2.7 CI 95% −4.1; −1.4, *p* < 0.001) and the FSS (Δ = −0.3% CI 95% −0.4; −0.1, *p* < 0.001). These findings demonstrate the importance of incorporating approaches into a BWRP that increase lower limb muscle power and decrease fatigue severity and thus reduce disability in adults with obesity.

## 1. Introduction

The prevalence rate of obesity continues to increase worldwide, despite it being preventable [1]. In 2016, overweight and obesity rates in the world adult population aged 18 and older were estimated at 39% and 13%, respectively [1]. Since 1975, the global prevalence of obesity has tripled, despite worldwide efforts to combat it. The negative consequences of obesity on various body systems are well-established. It can lead to increased risks in numerous health conditions, including musculoskeletal conditions, cardiovascular diseases, metabolic syndrome, gastrointestinal, and pulmonary conditions [1,2]. Obesity is closely associated with metabolic syndrome, physical inactivity, and energy imbalance. Exercise, diet, and behavioral changes play an important role in the prevention and management of obesity and obesity-related conditions [3]. There is, therefore, a need to develop carefully constructed body weight reduction programs (BWRP) and to investigate their effectiveness in reducing the adverse health consequences associated with obesity.

Obesity also appears to hinder mobility and functional ability, leading to an acceleration in disability [4,5]. The promotion of physical activity and the reduction of sedentary behavior deliver well-established health benefits, including reductions in body mass index (BMI) and waist circumference (WC), and improvements in physical function for people with obesity [6,7,8]. Promoting lifestyle forms of physical activity such as increased walking in addition to structured exercise appears to be more effective in reducing body weight in people with obesity than either activity form alone [6]. A randomized controlled trial of 93 elderly people with obesity showed that exercise, in combination with weight loss, improved physical function and frailty status more than either intervention alone [7]. Another randomized controlled trial of 439 overweight-to-obese postmenopausal sedentary women reported that a combination of a calorie-reduced, low-fat diet and a moderate-intensity, facility-based aerobic exercise program showed greater improvements in body weight and adiposity compared to either intervention alone in these participants [8]. Therefore, lifestyle changes involving exercise participation, together with a healthy diet, remain significant to body weight reduction, because an imbalance between energy intake and expenditure plays a crucial role in obesity. This emphasizes the importance of incorporating both diet and physical activity into a BWRP to improve weight loss, body composition, and physical function.

To date, research exploring the importance of performing an inpatient BWRP, specifically physical activity in hospitalized adults with obesity, is lacking. There is also a need to investigate how an inpatient BWRP affects the level of function (disability) in activities of daily living in patients with obesity, since research into this area is similarly scarce. Therefore, the primary aim of the present study was to explore the short-term changes in disability after an inpatient BWRP in adults with obesity. The secondary aim was to determine the association between changes in the level of disability and the health parameters of body mass, BMI, body composition, metabolic variables, blood pressure, fatigue severity, and lower limb muscle power after the BWRP in these patients.

## 2. Materials and Methods

The present study employed a pre–post study design to explore the short-term changes in the level of disability after the BWRP in patients with obesity.

### 2.1. Participants

Individuals with obesity (BMI > 35 kg/m^2^) were recruited at the Division of Metabolic Diseases, Istituto Auxologico Italiano, IRCCS, Piancavallo (VB), Italy where they were hospitalized. The patients performed a 3-week, multidisciplinary BWRP entailing nutritional education, an energy-restricted diet, psychological counseling, and physical rehabilitation (moderate aerobic activity), as previously described [9]. Metabolic syndrome was defined as the presence of three abnormal findings out of the following five parameters: central obesity, high systolic (SBP) and/or diastolic (DBP) blood pressure, high triglycerides, low high-density lipoprotein (HDL) cholesterol, or elevated fasting glucose [10]. Central obesity was defined as WC of ≥94 cm in men and ≥80 cm in women. Hypertension was defined as the presence of SBP values of >130 mmHg and/or DBP values of > 85 mmHg, or in the case of antihypertensive drugs use. Hypertriglyceridemia was defined as the presence of triglycerides values of >150 mg/dL, or in the case of a specific treatment. A low HDL cholesterol level was defined as values of <50 mg/dL. Elevated fasting glucose was defined as glycemia of >100 mg/dL, or in the case of antidiabetic drugs use.

The study was approved by the Ethics Committee of Istituto Auxologico Italiano, Milan, Italy (registration code: 2013_06_27; project code: 18A301) and was performed in accordance with the Helsinki Declaration of 1975, as revised in 2008. The purpose and objective of the study were explained to each participant and written informed consent was obtained from all eligible participants. Participants were excluded from the study if they had any difficulty in performing the functional test scheduled in the program (i.e., orthopedic, psychiatric, and neurological disorders), or if they were taking drugs known to interfere with the tests (i.e., psychotropics).

### 2.2. Body Weight Reduction Program

Individualized diets were provided to each participant during the 3-week BWRP. The amount of energy given in the diet was calculated by subtracting approximately 500 kcal from the measurement of resting energy expenditure. The diet, in terms of macronutrients, contained 21% proteins, 53% carbohydrates, and 26% lipids; the daily estimated water content was 1000 mL, while the estimated salt content was 1560 mg Na+, 3600 mg K+, and 900 mg Ca+2. Extra water intake of at least 2000 mL/day was encouraged [9]. Indirect calorimetry (Vmax 29; SensorMedics Corporation, Yorba Linda, CA, USA) was used to determine the value of resting energy for 20 min for each participant. Participants could choose from a daily menu containing a variety of foods. Foods to which participants reported to be allergic were removed from the menu. Five portions of fruits and vegetables per day were compulsory.

Nutritional education, comprised of lectures, demonstrations, as well as group discussions, was provided daily to all participants throughout the period of the program. Psychological counseling with regard to cognitive–behavioral strategies, including problem solving and stress management, development of healthy eating habits and cognitive restructuring, was delivered 2–3 times per week by clinical psychologists.

All participants engaged in an individualized daily exercise program of 45 to 60 min with heart rate (HR) monitoring for five days per week (weekdays) under the supervision of a qualified therapist. The physical activity program consisted of five days per week training and included: (i) one hour of dynamic aerobic standing and floor exercises with arms and legs, at moderate intensity, and under the guidance of a therapist; (ii) either 20–30 min of cycle ergometer exercise at 60 W, or 3–4 km of outdoor walking on flat terrain, according to individual capabilities and clinical status. In addition, subjects had one hour per day of aerobic-free activities at the institution on Saturdays and Sundays. They were asked to complete at least 95% of the exercise program, the completion of which was verified by the research assistant and the physical trainers.

### 2.3. Anthropometric Measurements

The height and weight of participants were measured using a weight scale with a stadiometer (Wunder Sa.Bi., WU150, Trezzo sull’Adda, Italy). WC in standing was measured with a flexible tape placed halfway between the inferior margin of the ribs and the superior border of the iliac crest. Body composition was measured by means of a multifrequency tetrapolar impedance meter (BIA, Human-IM Scan, DS-Medigroup, Milan, Italy) with a delivered current of 800 μA at a frequency of 50 kHz. In order to reduce errors of measurement, special attention was paid to the standardization of the variables that were known to affect measurement validity, reproducibility, and precision. Measurements were performed according to the method of Lukaski [11] (after 20 min resting in a supine position with arms and legs relaxed and not in contact with other body parts) and under strictly controlled environmental conditions.

### 2.4. Metabolic Variables

Lipids (HDL cholesterol and triglycerides) and glucose were obtained from a total of 12 mL blood samples collected at the beginning of the BWRP. Serum glucose level was determined by the glucose oxidase enzymatic method (Roche Diagnostics, Monza, Italy) with a sensitivity of 2 mg/dL, whilst serum HDL cholesterol and triglycerides levels were measured by colorimetric enzymatic assays (Roche Diagnostics, Monza, Italy) with sensitivities of 3.09 mg/dL and 8.85 mg/dL, respectively.

### 2.5. Blood Pressure

Blood pressure was determined using a sphygmomanometer placed on the right arm of participants in a seated position. Recorded blood pressure was the average of three measurements taken 10 min apart for SBP and DBP.

### 2.6. Functional Test

The stair climbing test (SCT), commonly applied to assess lower limb muscle power and functional ability, was used as the functional test in the current study [12,13]. Previous studies have also used this test to evaluate anaerobic muscle power in adults with obesity [14,15]. Participants were asked to climb a 1.99 m high staircase containing 13 steps, each of 15.3 cm, at their maximum speed. Each participant was allowed one to two trials to familiarize themselves with the test procedures prior to performing the experimental test considered in the analysis. The total time taken to perform the test trial was measured with a digital watch.

### 2.7. Fatigue Severity Scale (FSS)

The FSS is a self-reported questionnaire that is broadly applied to assess the level of fatigue associated with a wide range of chronic diseases [16,17]. Previous studies conducted in patients with obesity have used this scale to measure the severity of fatigue and it has been validated in Italian individuals [18,19]. The scale comprises of nine items that assess the negative effects of fatigue on a person’s activities and lifestyle, using a Likert-like scale with scores ranging from 1 (strong disagreement) to 7 (strong agreement). The total score is calculated as the sum of the scores from the nine items.

### 2.8. Disability Level

The Oswestry disability index (ODI), which is a valid and vigorous measure of patient-reported disability [20], was applied to assess participants’ disability and assign them to one of three disability groups, namely, minimal (0–20% ODI), moderate (21–40% ODI) and severe (41–80% ODI). The ODI is composed of 10 items associated with function in activities of daily living, including pain intensity, personal care, lifting, walking, sitting, standing, sex life, social life, and travelling. Patients are asked to choose one out of six statements (items), with scores ranging from 0 to 5 for each item, based on their ability to manage these daily activities, whilst taking pain into account. The total score for all items is then divided by the total possible score and multiplied by 100 to provide a percentage of disability. Therefore, the total score of ODI ranges between 0 (no disability) and 100 (maximum disability). In the present study, only 7 out of the 10 ODI items were used to determine disability, since the last 3 items were not pertinent to hospitalized patients. A minimum improvement in the total ODI score of 10 points was considered to be a minimum clinically important change [21].

### 2.9. Statistical Analysis

Descriptive statistics and inferential analyses were performed using R version 3.6.0 [22]. In the descriptive statistics, mean values and standard deviations (SD) for the participant characteristics of age, gender, BMI, body composition, metabolic variables, blood pressure, fatigue severity, lower limb muscle power, and the level of disability were determined. The Shapiro–Wilk test was applied to check for data normality. Analysis of variance (ANOVA) for normally distributed data and Kruskal–Wallis for non-normally distributed parameters were employed to compare baseline characteristics between the disability groups. The Chi-square test was used for categorical variables. An age-and-sex-adjusted two-way repeated measures ANOVA was used to analyze changes in the parameters between pre and post intervention. The ANOVA tests were followed by Tukey’s post hoc tests. Linear regression was used to determine the association between changes in the health parameters and the level of disability after the BWRP, and *p* values less than 0.05 were considered to be statistically significant.

## 3. Results

A total of 160 consecutive adults with obesity (males: 52, females: 108) were recruited into the study. Sixty (37.5%) suffered from metabolic syndrome (males: 14, females: 46, mean age: 51.0 ± 14.8 years, BMI: 44.6 ± 4.3 kg/m^2^, ODI: 19.0 ± 16.3%), while 100 were without metabolic syndrome (males: 38, females: 62, mean age: 48.7 ± 15.9 years, BMI: 42.5± 5.8 kg/m^2^, ODI: 18.5 ± 15.5%). The primary outcome variable “the level of disability measured by ODI” did not differ (χ2(2) = 0.02, *p* = 0.87) between obese participants with or without metabolic syndrome. The baseline characteristics of participants are presented in Table 1. The ODI scores were weakly, but significantly, correlated with body weight (Spearman’s r = −0.27), FFM (r = −0.24), HDL cholesterol (r = 0.18), and glycemia (r = 0.24) at baseline. SCT time (r = 0.56) and FSS (r = 0.51) scores were moderately correlated with ODI scores.

Participants’ characteristics of the three disability subgroups, namely minimal, moderate, and severe, are presented in Table 2.

### 3.1. Changes in the Level of Disability and Health Parameters after the BWRP

At the end of the intervention, all the parameters had improved, including body weight (Δ = −5.0 kg CI 95% −5.3; −4.6, *p* < 0.001), BMI (Δ = −1.8 kg/m^2^ CI 95% −1.9; −1.7, *p* < 0.001), FFM (Δ = 1.3% CI 95% 0.9; 1.8, *p* < 0.001), FM (Δ = −4.1 kg CI 95% −4.7; −3.5, *p* < 0.001), SBP (Δ = −8.9 mmHg CI 95% −11.3; −6.6, *p* < 0.001) and DBP (Δ = −4.0 mmHg CI 95% −5.5; −2.6, *p* < 0.001), HR (Δ = −6.6 CI 95% −8.6; −4.6, *p* < 0.001), SCT (−0.8 sec CI 95% −1.0; −0.6, *p* < 0.001), FSS (Δ = −12.0 CI 95% −13.5; −10.5, *p* < 0.001), and ODI (Δ = −7.2% CI 95% −8.6; −5.7, *p* < 0.001).

Regarding disability, Figure 1 shows the improvements in the parameters that were significantly different between the three disability groups. Reductions in body weight and FM were significantly greater in the minimal disability group than in the severe disability group (Figure 1a,c), whereas FFM and FSS improved more in individuals with moderate and severe disability, respectively, than those with minimal disability (Figure 1b,e). Clinically meaningful reductions in disability were observed in the moderate disability group (Δ = −11.8% CI 95% −14.3; −9.3, *p* < 0.001) and the severe disability group (Δ = −15.9% CI 95% −19.6; −12.2, *p* < 0.001) (Figure 1f).

### 3.2. The Association between Changes in Health Parameters and Disability (ODI) after the BWRP

Age-and-sex-adjusted linear regression models showed that improvements in the SCT (*t* = 4.0, *p* < 0.001) and FSS (*t* = 3.8, *p* < 0.001) explained the reductions in the ODI (disability) throughout the period of the intervention. The slope coefficient for the SCT was 2.7% (95% CI, 1.4%, 4.1%), meaning that ODI (disability) improved by 2.7% for each second improvement of the SCT, whilst the same amount of disability reduction was explained by every 10-point improvement of the FSS. Changes in the ODI components unrelated to stair climbing, such as personal care, pain intensity, lifting, standing, and sitting, were also individually explained by improvements in the SCT (*p* < 0.05). Changes in the other parameters were not associated with disability reduction (*p* > 0.05) (Figure 2).

## 4. Discussion

The purpose of the present study was to examine the short-term changes in disability after an inpatient BWRP in adults with obesity. An additional aim was to determine the association between changes in the level of disability and the health parameters of body mass, body mass index (BMI), body composition, metabolic variables, blood pressure, fatigue severity, and lower limb muscle power after the BWRP in these patients. Over the period of the intervention, all these variables showed an improvement in at least one of the three disability groups of patients with obesity. However, reductions in disability were explained only by the improvements in lower limb muscle power and fatigue severity, measured by the SCT and the FSS, respectively. Additionally, the ODI of patients with moderate and severe disability at baseline demonstrated clinically meaningful improvements compared to individuals with minimal disability.

BMI, body composition, metabolic, and cardiovascular variables were not associated with disability reduction in patients with obesity. However, all these parameters showed an improvement after the BWRP. These findings are in line with previous studies conducted in both children and adults with obesity involving BWRPs [23,24,25]. However, changes to the parameters after the intervention were independent of the severity of the disability groups. Although some of these parameters, such as FFM, HDL cholesterol, and glycemia, were weakly correlated with disability at baseline, reductions in disability were not associated with the baseline metabolic variables or improvements in body composition, anthropometric, and cardiovascular parameters throughout the period of the three weeks. The improvements in BMI, body composition, and cardiovascular variables after our 3-week BWRP (determining a mean body weight reduction of 4.2%) may not actually be sufficient to influence disability of individuals suffering from severe obesity, which could benefit from more marked weight reduction (i.e., 5–10%, as described in previous studies) [26,27]. For this reason, additional studies with more prolonged periods of BWRP (in-hospital and out-hospital), determining greater body weight reduction, will be necessary to determine more significant changes in the level of disability.

The improvements in lower limb muscle power and fatigue severity, measured by the SCT and FSS, respectively, were the only parameters to explain a reduction in disability in patients with obesity. Of all the variables defining the basal characteristics of participants, the SCT and FSS were the only parameters moderately correlated with disability at baseline. Furthermore, each one second of improvement in the SCT resulted in a 2.7% reduction in disability. An equal amount of disability reduction was explained by every 10-point improvement in the FSS. The improvements in these parameters after the BWRP also paralleled the severity of disability and contributed to explaining the reductions in disability. One may argue that the relationship between an improvement in stair climbing ability and a reduction in disability is clear, since walking is one of the 10 measures of disability. Nevertheless, the improvement in the SCT was not only associated with reductions in the total ODI score, but also significantly correlated with improvements in the other components of the ODI, such as personal care, pain intensity, lifting, standing, and sitting. This implies that the improvement in the SCT score made a significant positive contribution to both the physical and psychological aspects of disability. A previous study of 118 patients with chronic low back pain undergoing lumbar fusion surgery found that one-minute stair climbing was correlated (r = 0.39, *p* < 0.01) with the ODI [28], which was a similar result to the correlation between the baseline SCT and ODI found in the current study. Additionally, a randomized controlled trial of five months duration with 126 elderly individuals with obesity showed that self-reported disability was improved more in the resistance training group with a weight-loss intervention than without a weight-loss intervention [29]. Although the nature of the intervention applied in this randomized controlled trial was different to that in the current study, weight reduction appears to contribute to a reduction in disability in the long term, at least in older adults. However, to date, studies examining the effects of a BWRP on BMI, body composition, metabolic, and cardiovascular parameters in relation to disability in people with obesity are lacking. Future studies are needed to investigate whether the positive effects on disability of the short-term BWRP can be maintained in the longer term. The findings from this current study demonstrate the importance of incorporating approaches in a BWRP that increase lower limb muscle power and lower fatigue severity to reduce disability in adults with obesity.

Disability across all three groups of patients with obesity was significantly improved from baseline to post-intervention. The BWRP resulted in clinically meaningful improvements in the moderate and severe disability groups, with reductions in the ODI scores of 11.8% (95% CI, −14.3; −9.3) and 15.9% (95% CI, −19.6; −12.2), respectively. The ODI score of the minimal disability group improved by 3.6% (95% CI, −5.2; −1.9), which was statistically significant but not clinically meaningful, suggesting that individuals with greater disability benefited more from the program than those with less disability. However, in the present study, these improvements were explained by only two of the parameters, namely the SCT and the FSS. Future studies are needed to specifically investigate the effects of BWRP in relation to disability and that explore other factors that could potentially contribute to reducing disability in individuals with obesity.

The main limitation of the current study was that the metabolic parameters were only measured at baseline, meaning that no information is available on whether they changed as a result of the intervention. Additionally, no disability subgroups were created for individuals with crippling and bed-bound disability, due to there being insufficient participants belonging to these groups. There is a requirement for future studies involving these groups to examine the effect of a BWRP on such disability. Only seven out of the ten ODI items were used to determine disability because the last three items (sexual life, social life, travelling) were not pertinent to hospitalized patients. This means that any association between these three items and the BWRP could not be ascertained.

In conclusion, the short-term, multidisciplinary BWRP significantly improved the level of disability among patients with obesity. From all the different parameters defining the characteristics of participants in the present study, reductions in disability were explained only by improvements in lower limb muscle power and fatigue severity. The findings from this study demonstrate the importance of incorporating approaches into a BWRP to increase lower limb muscle power and decrease fatigue severity, and thus to reduce disability in adults with obesity. Future studies are required to explore whether the positive effects of the short-term (three weeks) BWRP on disability can be maintained in the longer term, and to identify other factors that could potentially reduce disability in individuals with obesity.

## Figures and Tables

**Figure 1 jcm-11-03175-f001:**
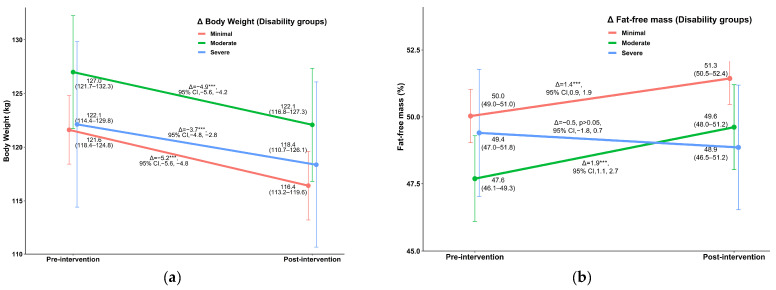
Improvements in outcome variables across the three disability groups (mean, standard error of the mean): (**a**) body weight loss; (**b**) fat-free mass; (**c**) fat mass; (**d**) stair climbing time (**e**) fatigue severity scale (**f**) Oswestry disability index. *** *p* < 0.001; CI—confidence interval; mean values (standard error of the mean); Δ—changes between pre and post intervention; FSS—fatigue severity scale; ODI—Oswestry disability index; an age-and-gender-adjusted two-way repeated measures ANOVA was used to determine changes in the parameters between pre and post intervention.

**Figure 2 jcm-11-03175-f002:**
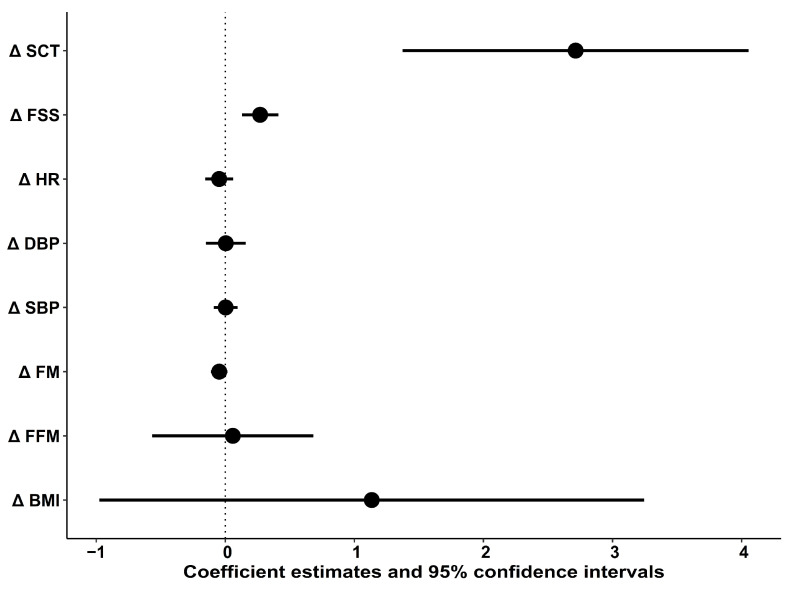
Associations between improvements in the parameters and reductions in ODI. Δ—changes between pre and post intervention; SCT—stair climb test; FSS—fatigue severity scale; HR—heart rate; DBP—diastolic blood pressure; SBP—systolic blood pressure; FM—fat mass; FFM—fat-free mass; BMI—body mass index.

**Table 1 jcm-11-03175-t001:** Participants’ characteristics at baseline (mean ± SD).

Variables	Baseline Characteristics (n = 160)
Age (years)	50.1 (15.2)
Sex (female)	67.5%
Body weight (kg)	118.9 (19.9)
BMI (kg/m^2^)	43.8 (5.4)
FFM (%)	47.9 (5.9)
FM (kg)	61.7 (13.6)
SBP (mmHg)	135.1 (14.8)
DBP (mmHg)	84.1 (8.5)
HR	83.0 (13.4)
SCT (sec)	5.5 (2.8)
FSS	38.5 (12.5)
Triglycerides (mg/dL)	140.4 (59.0)
HDL cholesterol (mg/dL)	49.8 (13.9)
Glycemia (mg/dL)	106.2 (23.6)
WC (cm)	122.3 (12.5)
ODI score (%)	18.7 (15.8)

SD—standard deviation; BMI—body mass index; FFM—fat-free mass; FM—fat mass; SBP—systolic blood pressure; DBP—diastolic blood pressure; HR—heart rate; SCT—stair climbing test; FSS—fatigue severity scale; HDL—high-density lipoprotein; WC—waist circumference; ODI—Oswestry disability index.

**Table 2 jcm-11-03175-t002:** Participants’ characteristics by disability groups at baseline (mean ± SD).

Variables	Groups Classified Based on ODI Score			
	Minimal Disability (*n* = 100)	Moderate Disability (*n* = 41)	Severe Disability (*n* = 19)	*p* Value
Age (years)	46.2 (15.8)	56.3 (11.2)	57.2 (12.9)	<0.001 ^k^
Sex (female)	55%	83%	100%	<0.001 ^c^
BMI (kg/m^2^)	43.2 (4.5)	45.5 (7.3)	43.6 (4.8)	0.41 ^k^
SCT (sec)	4.6 (1.6)	6.2 (2.0)	9.5 (4.8)	<0.001
FSS	34.2 (11.7)	43.2 (10.1)	50.5 (9.1)	<0.004 ^k^
Triglycerides (mg/dL)	141.0 (61.9)	139.0 (51.0)	140.0 (62.6)	0.88 ^k^
HDL cholesterol (mg/dL)	48.1 (12.7)	52.4 (16.7)	53.3 (12.9)	0.23 ^k^
Glycemia (mg/dL)	103.0 (23.9)	114.0 (21.1)	106.0 (24.9)	0.003 ^k^
WC (cm)	122.0 (11.8)	125.0 (15.0)	118.0 (8.5)	0.20 ^a^
METSYND	59%	78%	47%	0.04 ^c^

*p* value—statistical significance computed by using Kruskal–Wallis test ^k^, ANOVA ^a^, and Chi-square test ^c^ for comparison between the three groups; SD—standard deviation; BMI—body mass index; SCT—stair climbing test; FSS—fatigue severity scale; HDL—high-density lipoprotein; WC—waist circumference; METSYND—metabolic syndrome prevalence.

## Data Availability

The data presented in this study are available upon a reasonable request from the corresponding author.

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
