# Peer review of "Changes in the Oswestry Disability Index after a 3-Week In-Patient Multidisciplinary Body Weight Reduction Program in Adults with Obesity"

_jcm, 2022, doi:10.3390/jcm11113175_

Round 1

Reviewer 1 Report

Major:

Key issue with this paper is that the title implies that Metabolic Syndrome is a contributing factor to the outcomes of this program, yet none of the analysis evaluates this.

Furthermore, the introduction focuses on metabolic syndrome – which is largely irrelevant given lack of analysis.

Abstract should report changes in weight – and discussion should indicate whether these were clinically meaningful?

Figures. These are repetitive. Pre-post shows the same information as the change score figure.

But I feel the authors need to consider the axis and direction of change. As presented the change for body weight in figure 1B is positive – so +5.2kg…. Shouldn’t this be presented as a reduction so -5.2kg.
3 weeks – baseline. Same for other variables where the benefit should be a reduction e.g climbing time or fat mass.

Line 278 how was function in daily living evaluated?

Line 282 and 289 “…..regardless of Metabolic Syndrome status”. I don’t think you can make this conclusion - you have not tested this. All data were evaluated together

Line 344 A key limitation is that you were unable to determine whether met syndrome status changed as a result of the intervention.

The paper would benefit from a discussion on why changes in body weight did not relate to ODI.

Minor

Line 15 BMI needs units

Line 22-24 needs to include units for variable change scores

Line 61 should this be high risk LOW-density not high density

Line 78 Not convinced this is a case series design given the intervention

Line 89. WC cut offs differ for men

Line 94 glucose units in mmol/L (mg/dL) other bloods in mg/dL only – consistency

Line 135 need reference for the international guidelines re body comp

Line 170 how were the ODI cut-offs chosen?

Line 184-185 no mention of posture/hi motion etc in method or results/discussion

Table 1 and 2 need units

Results – throughout this entire section, data are presented without units in text or tables. E.g line 198 should be mean age: 48.7±15.9 years, BMI:42.5± 5.8 kg/m2

Author Response

Reviewer 1

Thank you for taking the time to review our article. Your feedback was very helpful in improving the overall quality of the manuscript. We have carefully considered and addressed every comment and have revised the manuscript accordingly. We believe that the changes made as a result of these suggestions have improved this manuscript significantly. Please see a point-by-point response to your suggested edits below.

Q1. Key issue with this paper is that the title implies that Metabolic Syndrome is a contributing factor to the outcomes of this program, yet none of the analysis evaluates this.

Furthermore, the introduction focuses on metabolic syndrome – which is largely irrelevant given lack of analysis.

A1. We agree with the reviewer that we have underlined excessively the presence (or not) of metabolic syndrome both in the title and in the introduction, despite its indeed irrilevant role in affecting the results as documented by our results. For this reason, we have modified the title as follows “Changes in the Oswestry Disability Index after a 3-weeks in-patient multidisciplinary body weight reduction program in adults with obesity”, and the introduction has now mainly focused on obesity. Discussion around metabolic syndrome in the introduction has been removed as suggested.

Lines (changes made): 1-4

Q2. Abstract should report changes in weight – and discussion should indicate whether these were clinically meaningful?

A2. This relevant information has been added.

Lines (changes made): 20,21

Q3. Figures. These are repetitive. Pre-post shows the same information as the change score figure.

A3. We agree with this specific comment. The “change score” figure has been removed as suggested.

Lines (changes made): 234-240

Q4.But I feel the authors need to consider the axis and direction of change. As presented the change for body weight in figure 1B is positive – so +5.2kg…. Shouldn’t this be presented as a reduction so -5.2kg.
3 weeks – baseline. Same for other variables where the benefit should be a reduction e.g climbing time or fat mass.

A4. We agree with this comment and have checked the numbers reported in the Figures. The direction and axis of change (values) at the “pre-post” figure have now been presented as suggested.

Lines (changes made): 234-240

Q5. Line 278 how was function in daily living evaluated?

A5. As ODI comprises of items associated with function in activities of daily living, we used “Function in daily living” as the representation of ODI in the first submission of the manuscript but the terminology has now been replaced with the direct expression of ODI “the level of disability” in order to avoid confusion.

Lines (changes made): 62, 264

Q6. Line 282 and 289 “…..regardless of Metabolic Syndrome status”. I don’t think you can make this conclusion - you have not tested this. All data were evaluated together

A6. The sentence has been modified and the phrase “regardless of metabolic syndrome status” has been deleted.

Lines (changes made): 21, 44, 268, 275, 292, 342

Q7. Line 344 A key limitation is that you were unable to determine whether met syndrome status changed as a result of the intervention.

A7. The evaluation of changes in metabolic syndrome prevalence after 3-week BWRP was actually out of the aim of the present study. Since no differences were recorded between patients with or without metabolic syndrome in basal condition, the evaluation of changes in metabolic syndrome prevalence seemed actually scarcely relevant. The main limitation has now been modified as suggested.

Lines (changes made): 332-334

Q8. The paper would benefit from a discussion on why changes in body weight did not relate to ODI.

A8. This point has now been included in the discussion section.

Lines (changes made): 285-289

Minor

Q9. Line 15 BMI needs units

A9. Units have been added to the text and tables.

Lines (changes made): 16, 208-209

Q10.Line 22-24 needs to include units for variable change scores

A10. Units have been added to the text.

Lines (changes made): 20-24

Q11. Line 61 should this be high risk LOW-density not high density

A11. It has been corrected.

Lines (changes made): 51

Q12. Line 78 Not convinced this is a case series design given the intervention

A12. It has been changed to “a pre-post study design”

Lines (changes made): 67

Q13. Line 89. WC cut offs differ for men

A13. As reported in ref. 3, WC cut off for males was actually > 94 cm. The cut-off value of 80 cm has been corrected with 94 cm in the revised manuscript.

Lines (changes made): 78

Q14. Line 94 glucose units in mmol/L (mg/dL) other bloods in mg/dL only – consistency

A14. We have chosen to use only mg/dl

Lines (changes made): 81-83

Q15. Line 135 need reference for the international guidelines re body comp

A15. We have updated the text and referenced the international guidelines.

Lines (changes made): 124-131

Q16. Line 170 how were the ODI cut-offs chosen?

A16. We have classified participants according to the ODI scoring tool, where the minimal disability group includes patients with ODI scores ranging 0-20%. This has been clarified in the text.

Lines (changes made): 163, 164

Q17. Line 184-185 no mention of posture/hi motion etc in method or results/discussion

A17. Posture/motion characteristics not relevant to the results have been removed.

Lines (changes made): 179, 180

Q18. Table 1 and 2 need units

A18. The tables have been updated with units.

Lines (changes made): 201-202, 208-209

Q19. Results – throughout this entire section, data are presented without units in text or tables. E.g line 198 should be mean age: 48.7±15.9 years, BMI:42.5± 5.8 kg/m2

A19. The text and tables have now been updated with units.

Lines (changes made): 20-24, 193, 195, 201, 202, 208, 209, 214-222

Reviewer 2 Report

I would like to thank you for the opportunity to review this manuscript.

This is an interesting manuscript, on a relevant topic, yet several issues should be addressed to improve its’ quality.

General:

  • Although the manuscript is overall well-written, it should be revised by a native speaker. Several grammatical issues may be found across the manuscript.
  • "People-first" language should be used in the present manuscript. A person should not be defined by his/her condition. Please refer to "the person with obesity" and not "the obese person". Please revise the manuscript accordingly, including the title.
  • It is not clear how the authors may prove that the changes in the investigated parameters were induced by the program, as suggested by the title (“effect”) and stated in the introduction section (p.2, lines 74-76), without a control group (in-patients with obesity not exposed to the program).
  • There are also some concerns regarding the methods used. For instance, despite the given reference, short, but relevant information about the physical activity sessions should be included in the present manuscript, including how physical activity intensity was measured. Also, It is not clear where the authors have supported the use of a cut-off value of >80 cm for waist circumference (for both men and women) since it is not in line with the cited reference, or (to the best of my knowledge) with any other reference or recommendation.
  • The Results section has confusing headings. I think that the results section would benefit from restructuring.
  • Also unclear is the fact that the authors discuss unreported data (p10, lines 313-317).

Specific issues:

  • Abstract, p.1, lines 14-15: “Abstract: To examine…” – As is, it is not clear that this is the study's aim.
  • Introduction, p.2, lines 59-60: The sentence seems redundant. Please rephrase.
  • Introduction, p.2, line 71: “(carefully designed) BWRP” - It is not clear if video calls were also used. All ways of communication should be reported.
  • Methods, p.3, lines 107-109: This sentence seems inconsistent with the previous one, in which the authors state that diets were (also) based on "the baseline physical activity level."
  • Methods, p.3, line 126: “ergometer” – do the authors intend to say “cycle ergometer”?
  • Methods, p.3, line 132: “scale” – do the authors intend to say “weight scale”?
  • Methods, p.3, line 135: “crista” – do the authors intend to say “Iliac crest”?
  • Methods, p.4, lines 156-157: Please rephrase using appropriate terms. As is it seems confusing (two trials followed by one trial).
  • Methods, p.4, lines 164-165: Please revise. It is not common to use a scale that does not vary.
  • Methods, p.4, line 170: “(20<ODI=40%)” - I suggest "(20-40% ODI)". Please revise the other categories. Reading is hard as the categories are presented.
  • Results, p.5, line 198: “16.3, while…” - Parenthesis missing.
  • Results, p.6, line 225: “improved among the obese patients” - Were any other patients included, besides those with obesity?
  • Results, p.6, lines 234-235: The identified sentence seems a repetition of the one in lines 225-226.
  • Results, p.8, lines 263-264: Was not ODI the dependent variable? The Writing should be consistent with the theoretical constructs... in other words, improvements in SCT and FSS may explain ODI improvement...
  • Discussion, p.9, line 285: “ODI of patients with moderate and severe disability demonstrated” – I suggest: “moderate and severe disability at baseline”.
  • Table 1: Table 1 title is unsuitable based on the information presented. In fact, Table 1 would benefit from the inclusion of the percentage of women/men per group (with and without MS). Also, the inclusion of correlation results mixed with participants' baseline characteristics, does not improve reading or comprehension.

Author Response

Reviewer 2

I would like to thank you for the opportunity to review this manuscript.

This is an interesting manuscript, on a relevant topic, yet several issues should be addressed to improve its’ quality.

Thank you for taking the time to review our article. Your feedback was very helpful in improving the overall quality of the manuscript. We have carefully considered and addressed every comment and have revised the manuscript accordingly. We believe that the changes made as a result of these suggestions have improved this manuscript significantly. Please see a point-by-point response to your suggested edits below.

General:

Q1. Although the manuscript is overall well-written, it should be revised by a native speaker. Several grammatical issues may be found across the manuscript.

A1. As suggested, the manuscript has been proofread and edited by a native English speaker from professional proofreading and editing services. We believe that sentence construction for clarity, flow, readability, and grammar of the manuscript have been improved significantly.

Q2. "People-first" language should be used in the present manuscript. A person should not be defined by his/her condition. Please refer to "the person with obesity" and not "the obese person". Please revise the manuscript accordingly, including the title.

A2. Thank you for the suggestion. The terms have now been replaced with “People-first” language throughout the manuscript.

Lines (changes made): 15, 27, 45, 57, 59, 61, 68, 70, 146, 155, 191, 263, 267, 274, 276, 292, 309, 316, 320, 331, 342, 347, 350

Q3. It is not clear how the authors may prove that the changes in the investigated parameters were induced by the program, as suggested by the title (“effect”) and stated in the introduction section (p.2, lines 74-76), without a control group (in-patients with obesity not exposed to the program).

A3. The observation of the reviewer is actually correct. However, as the reviewer can obviously understand we have not the opportunity to evaluate a control group of in-patients with obesity not exposed to the program. The title and aim of the study have been changed as follows:

The title – “Changes in the Oswestry Disability Index after a 3-weeks in-patient multidisciplinary body weight reduction program in adults with obesity”

The aim – “the primary aim of the present study was to explore the short-term changes in disability after an inpatient BWRP in adults with obesity”, thus relieving the impact of the original title.

We have also replaced the phrase “induced by the intervention or program” by “after the intervention” in the revised manuscript. The word “effect” has been removed from the title and aim of the present study.

Lines (changes made): 1-3, 14, 60, 61, 64, 67, 68, 188, 213, 246, 262, 263, 266, 277

Q4. There are also some concerns regarding the methods used. For instance, despite the given reference, short, but relevant information about the physical activity sessions should be included in the present manuscript, including how physical activity intensity was measured.

A4. More information in relation to physical activity has been added in the revised version.  

Lines (changes made): 112-118

Q5. Also, It is not clear where the authors have supported the use of a cut-off value of >80 cm for waist circumference (for both men and women) since it is not in line with the cited reference, or (to the best of my knowledge) with any other reference or recommendation.

A5. As reported in ref. 13, WC cut off for males was actually > 94 cm. The cut-off value of 80 cm has been corrected with 94 cm in the revised manuscript.

Lines (changes made): 78

Q6. The Results section has confusing headings. I think that the results section would benefit from restructuring.

A6. The results section has been restructured and the headings have also been modified as suggested.

Lines (changes made): 190-255

Q7. Also unclear is the fact that the authors discuss unreported data (p10, lines 313-317).ù

A7. The associations in relation to the ODI components, previously missing, have been added to the results section.

Lines (changes made): 252-254

Specific issues:

Q8. Abstract, p.1, lines 14-15: “Abstract: To examine…” – As is, it is not clear that this is the study's aim.

A8. “To examine…..” has now been rewritten to clarify the aim as follows

“The aim of this study was to examine” 

Lines (changes made): 14

Q9. Introduction, p.2, lines 59-60: The sentence seems redundant. Please rephrase.

A9. The sentence has been rephrased.

Lines (changes made): 49-52

Q10. Introduction, p.2, line 71: “(carefully designed) BWRP” - It is not clear if video calls were also used. All ways of communication should be reported.

A10. The question raised by the reviewer is not clear to us. The patients were hospitalized during our 3-week in-hospital BWRP. No video calls were scheduled during the hospitalization, since the patients were directly attended (and monitored) by the multidisciplinary team, depending on the different specific activities scheduled in the patients’ daily program (as described in section 2.2). 

Q11. Methods, p.3, lines 107-109: This sentence seems inconsistent with the previous one, in which the authors state that diets were (also) based on "the baseline physical activity level."

A11. The sentences have been changed as follows “Individualized diets were provided to each participant during the 3-week BWRP. The amount of energy given in the diet was calculated by subtracting approximately…..”

Lines (changes made): 94, 95

Q12. Methods, p.3, line 126: “ergometer” – do the authors intend to say “cycle ergometer”?

A12. “ergometer” has been changed to “cycle ergometer” in the revised manuscript.

Lines (changes made): 115

Q13. Methods, p.3, line 132: “scale” – do the authors intend to say “weight scale”?

A13. “scale” has been changed to “weight scale”.

Lines (changes made): 121

Q14. Methods, p.3, line 135: “crista” – do the authors intend to say “Iliac crest”?

A14. “crista” has been changed to “Iliac crest”.

Lines (changes made): 124

Q15. Methods, p.4, lines 156-157: Please rephrase using appropriate terms. As is it seems confusing (two trials followed by one trial).

A15. In order to avoid misunderstanding, the sentence has been changed as follows: “Each participant was allowed one to two trials to familiarize themselves with the test procedures prior to performing the experimental test considered in the analysis”

Lines (changes made): 148-150

Q16. Methods, p.4, lines 164-165: Please revise. It is not common to use a scale that does not vary.

A16. Thank you for your correction. The sentence has been corrected as follows “…ranging from 1 (strong disagreement) to 7 (strong agreement)”

Lines (changes made): 158

Q17. Methods, p.4, line 170: “(20<ODI=40%)” - I suggest "(20-40% ODI)". Please revise the other categories. Reading is hard as the categories are presented.

A17. The format has been changed as suggested.

Lines (changes made): 163, 164

Q18. Results, p.5, line 198: “16.3, while…” - Parenthesis missing.

A18. Parenthesis has been added.

Lines (changes made): 193

Q19. Results, p.6, line 225: “improved among the obese patients” - Were any other patients included, besides those with obesity?

A19. The sentence has been changed as follows: “At the end of the intervention, all the parameters had improved, including body weight………”

Lines (changes made): 214

Q20. Results, p.6, lines 234-235: The identified sentence seems a repetition of the one in lines 225-226.

A20. The sentence has been removed.

Lines (changes made): 229

Q21. Results, p.8, lines 263-264: Was not ODI the dependent variable? The Writing should be consistent with the theoretical constructs... in other words, improvements in SCT and FSS may explain ODI improvement...

A21. The relevant sentences have been modified as suggested.

Lines (changes made): 24, 223, 224, 247-248, 268, 269, 343, 344

Q22. Discussion, p.9, line 285: “ODI of patients with moderate and severe disability demonstrated” – I suggest: “moderate and severe disability at baseline”.

A22. The sentence has been changed as suggested.

Lines (changes made): 271

Q23. Table 1: Table 1 title is unsuitable based on the information presented. In fact, Table 1 would benefit from the inclusion of the percentage of women/men per group (with and without MS). Also, the inclusion of correlation results mixed with participants' baseline characteristics, does not improve reading or comprehension.

A23. As suggested by the Reviewer 1, the lack of baseline significant differences between patients with or without metabolic syndrome makes the comparisons less relevant. For this reason, we have considered the entire study group without making a distinction between patients with or without metabolic syndrome. The correlations between ODI and the parameters have been removed from the Table 1 as suggested.

Lines (changes made): 201-202

Round 2

Author Response

Reviewer 1

Thank you for taking the time to review our article. We have addressed every comment and have revised the manuscript accordingly. Please see a point-by-point response to your suggested edits below.

Q1. Key words still include metabolic syndrome.

A1.“Metabolic syndrome” has been removed from the key words. Lines (changes made): 28

Q2. Line 42 states “… need to develop carefully constructed body weight reduction programs (BWRP) and to investigate their effectiveness in halting the progression of metabolic syndrome.” This is irrelevant to this study as you do not present changes in metsyn due to BWRP

Paragraph beginning line 43 leads with effect of obesity on function YET the rest of the paragraph talks about how PA and lifestyle change modifies met syn and related metabolic variables. This paragraph should be talking about studies to improve function and disability through increasing lifestyle – currently it is misleading and irrelevant as you do not assess changes in most of the parameters mentioned in this paragraph

A2. As suggested, the sentence “……to investigate their effectiveness in halting the progression of metabolic syndrome.” has been modified into “……to investigate their effectiveness in reducing the adverse health consequences associated with obesity.” The term “metabolic syndrome” has been removed, as suggested. Lines (changes made): 42, 43

The paragraph has now excluded the content related to metabolic syndrome and talked about lifestyle changes involving physical activity and diet in relation to body weight loss and physical function. Lines (changes made): 45-57, 60, 61

Q3. There is no discussion about whether the amount lost is clinically meaningful (usually 5- 10%) – this also relates to Q8 – quick calc and less than 5% BW reduction

So may be that greater weight loss is required to induce meaningful changes in disability?

A3. We have now discussed specifically about clinically meaningful body weight reduction in relation to the body weight loss found in the present study as follows:

The improvements in BMI, body composition and cardiovascular variables after our 3-week BWRP (determining a mean body weight reduction of 4.2%) may not actually be sufficient to influence disability of individuals suffering from severe obesity, which it could benefit from more marked weight reduction (i.e. 5-10%, as described in a previous study (31.1). For this reason, additional studies with more prolonged periods of BWRP (in-hospital and out-hospital), determining greater body weight reduction, will be necessary to determine more significant changes in the level of disability.

Lines (changes made): 285-291

Q4. There are different cut-offs for men vs women. You need to state specific ones for men and women (i.e. two values in text) not only men or only women

A.4. We have added the values for both men and women as follows: waist circumference (WC) of ≥ 94 cm in men and ≥ 80 cm in women

Lines (changes made): 84, 85

ADDITIONAL POINTS:

Q5. line 70 and line 88 are essentially repeated. Put BMI criteria on line 70 and cut line 88  

A5. Changes have been made as suggested. Lines (changes made): 76

Q6. Line 131 strictly controlled ?environmental conditions”

A6. The word “environmental” has been added. Lines (changes made): 137

Q7. Line 200 r= 0.51

A7. “r=” has been added. Lines (changes made): 206

Q8. Table 1 data are mean ± SD? Need to state in table

A8. “mean ± SD” has been added. Lines (changes made): 207

Q9. Table 2 METSYN ?prevalence

A9. The word “prevalence” has been added. Lines (changes made): 218

Reviewer 2 Report

I consider my previous concerns about the manuscript appropriately addressed. The manuscript has been greatly improved. In my opinion, the manuscript is now suitable for publication, as is.

Well done!

Author Response

(The authors gave the same response as above.)
